# GAN-based Method for Synthesizing Multi-Focus Cell Images [*]

Ken'ich Morooka[1], Xueru Zhang[1], Shoko Miyauchi[1],
Ryo Kurazume[1], and Eiji Ohno[2]

[1] Graduate School of Information Science and Electrical Engineering
Kyushu University, Fukuoka 819-0395, Japan `morooka@ait.kyushu-u.ac.jp`
[2] Faculty of Health Sciences, Kyoto Tachibana University, Kyoto 607-8175, Japan

**Abstract.** This paper presents a method for synthesizing multi-focus cell images by using generative adversarial networks (GANs). The proposed method, called multi-focus image GAN (MI-GAN), consists of two generators. A base image generator synthesizes a 2D base cell image from random noise. Using the generated base image, a multi-focus cell image generator produces 11 realistic multi-focus images of the cell while considering the relationships between the images acquired at successive focus points. From experimental results, MI-GAN achieves the good performance to generate realistic multi-focus cell images.

**Keywords:** Multi-focus pathological images · GAN · Image synthesis.

## 1 Introduction

Cervical cancer screening is useful for early detection of cancers with less invasive natures. In the screening, cytotechnologists observe a tissue sample taken out from human body, and find pre-cancerous and cancer cells from the sample. Generally, one sample includes tens of thousands of cells. Among them, the number of cancer cells is much smaller than that of normal cells. Moreover, in the case of cervical cancer screening in Japan, only 120 of every 10,000 people may carry cancer cells, and 7 of them will be diagnosed as suffering from cancer. Owing to these, the detection of cancer cells is a hard and time-consuming task.

Recently, instead of the sample, whole slide images (WSIs) have become a common method for not only cancer screening but also another clinical applications [1]. WSIs are high resolution digital images with gigapixels acquired by scanning the enter sample and varying focus points. The use of WSIs enables to computerize the cancer screening. By applying image processing techniques, WSI has the potential to improve the accuracy and efficiency of the cancer screening including web-based remote diagnosis.

Now, we have been developing an automatic system of cervical cancer screening using WSI. The construction of the system needs many WSIs including cancer

[*] Supported by JST CREST Grant Number JPMJCR1786 and JSPS KAKENHI Grant Number JP19H04139, Japan.

and normal cells. However, as stated above, cancer cell images are difficult to collect compared with the case of normal cells. Therefore, there is a serious problem of the imbalance between normal and cancer cell images. The data imbalance makes it difficult to construct the system with acceptable accuracy.

Here, generative adversarial networks (GANs) [2] have achieved great success at generating realistic images. Recent researches [3–7] have developed GAN-based methods for pathological images. Hou et al. [3] applied GAN to synthesize image patches to generate large-scale histopathological images by integrating the patches. Hu et al. [4] proposed a GAN-based unsupervised learning of the visual attributions of cells. Another GAN application to pathological images is a stain normalization of the images. One challenge of using pathological images is their color or stain variations. To overcome the problem, GAN-based stain normalization methods [5–7] have been developed to transfer the stain style of a microscopic image into another one.

Most of GAN-based methods have focused on single-focus images including natural and pathological images. On the contrary, considering the WSI generation, WSI is also regarded as a sequence of multi-focus cytopathological images acquired at different focus. However, there are few GAN-based methods whose targets are the multi-focus images.

When the multi-focus images is regarded as an image sequence, the generation of multi-focus images is related to realistic video generation [8–10]. Generally, the aim of the video generation is to capture the changes of the appearance and motion of a target. On the contrary, in our case, GAN needs to learn the appearance changes of cells by varying a focus setting. This difference makes it difficult to apply previous GANs for video generation to the synthesis of multi-focus images.

In this paper, we propose a new GAN-based method, called multi-focus image GAN (MI-GAN), for synthesizing multi-focus cell images to construct virtual WSIs. MI-GAN is composed of two phases. The first phase is to from random noise, produce a base cell image which is in focus in the multi-focus images. In the second phase, MI-GAN produces realistic multi-focus images of the cell considering the relationships between the images acquired at successive focus points.

## 2    Method

Fig. 1 shows the architecture of our proposed MI-GAN system which generates a sequence of 11 multi-focus images of a cell. The size of each generated image is $64 \times 64$[pixel]. Here, we denote $I_{-5}, \ldots, I_{+5}$ as the 11 images. Especially, $I_0$ is a base cell image which is in focus in the multi-focus images.

The MI-GAN consists of two generators. A base image generator $G_1$ synthesizes a 2D base cell image $I_0$ from random noise. Using the generated base image $I_0$, a multi-focus cell image generator $G_2$ produces 11 realistic multi-focus images of the cell. The two generators are trained independently. In the following, we explain the architectures and training of the two generators.

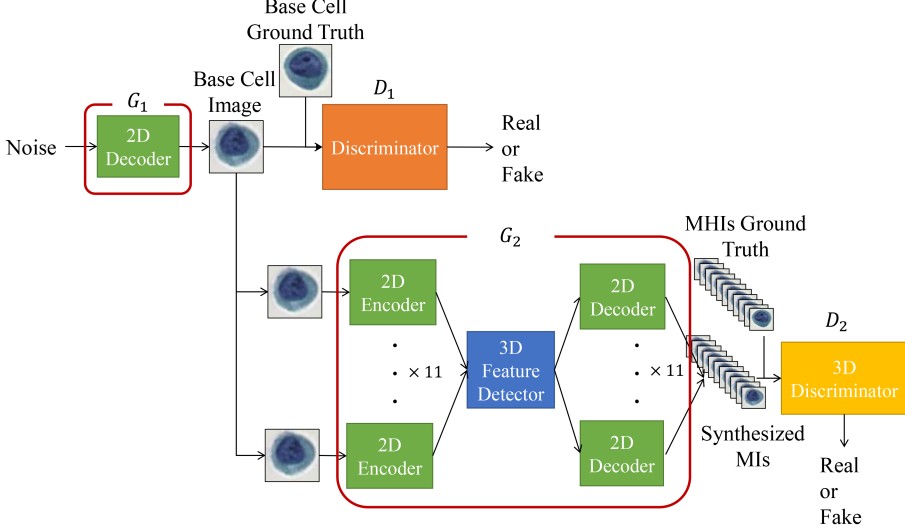

**Fig. 1.** Architecture of MI-GAN.

### 2.1 Base image generator

As shown in Fig. 2, the framework of constructing the base image generator is based on DCGAN [11]. Given a 100 dimensional random noise $Z$, the base image generator outputs the base image with size $64 \times 64$ [pixel]. In the base image generator, there are four up-sampling convolution layers. Batch normalization and Rectified Liner Unit (ReLU) activation are applied after each convolution layer. Moreover, the kernel size of the convolution layer is $6 \times 6$ while both the stride and padding sizes are 2. The discriminator is a feed-forward network with six convolution layers. The kernel size of the convolution layer is $5 \times 5$.

In the training of the base image generator, we use the loss function used in WGAN-GP [12] to stably synthesize images with acceptable quality. In the WGAN-GP, the loss function $\mathcal{L}_G^{(1)}$ of the generator $G_1$ is defined by

$$\mathcal{L}_G^{(1)} = -\mathbb{E}_{I \sim P_g}[D_1(I)]. \tag{1}$$

On the contrary, the loss function $\mathcal{L}_D^{(1)}$ of the discriminator $D_1$ is formulated as

$$\mathcal{L}_D^{(1)} = \mathbb{E}_{I \sim P_g}[D_1(I)] - \mathbb{E}_{I^* \sim P_r}[D_1(I^*)] + \lambda_1 \mathbb{E}_{\hat{I} \sim p_{\hat{I}}}[(\|\nabla_{\hat{I}} D_1(\hat{I})\|_2 - 1)^2] \tag{2}$$

where $I^*$ and $I$ are the real and synthesized base images of cells. The value of $\lambda_1$ in our method is set to 10. In eq. (2), $\hat{I}$ is calculated by

$$\hat{I} = \epsilon_1 I^* + (1 - \epsilon_1)I \tag{3}$$

where $\epsilon_1$ is a random number follow $U \sim [0, 1]$.

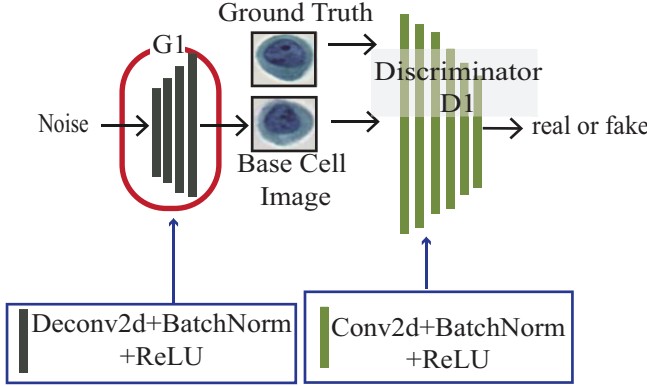

**Fig. 2.** Architecture of base image generation network.

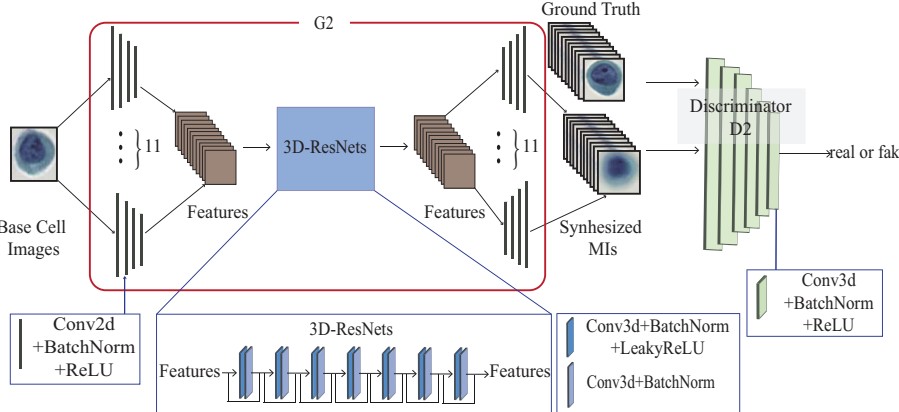

**Fig. 3.** Architecture of multi-focus cell image generation network.

## 2.2   Multi-focus cell image generator

Unlike general generators using random noise as the input data, the multi-focus cell image generator $G_2$ produces 10 multi-focus cell images from the base image. Here, cycleGAN [13] converts a given real image into another type image. Inspired by cycleGAN, as shown in Fig. 3, the multi-focus cell image generator is composed of three parts: an 2D encoder, a 3D feature map generation, and a 2D decoder.

The encoder part includes 11 networks, each of which uses the base image $I_0$ as the input image to output a candidate sample of the cell image acquired at the corresponding focus point. The input matrix of the feature map generation part is obtained by concatenating 11 candidate samples extracted from the 11 networks of the encoder part. The size of the input matrix is $11 \times 3 \times 4 \times 4$. 3D ResNet with seven layers is employed to transform the input matrix into

a 3D feature map of a sequence of multi-focus images while considering the relationships between the images acquired at successive focus points. In the decoder part, the feature matrix is divided into 11 2D feature maps with size of $3 \times 4 \times 4$. Each 2D feature map is inputted to the corresponding 2D-decoder part to synthesize the multi-focus cell image.

In the encoder part, there are four 2D down-sampling convolution layers. On the contrary, the decoder part has four 2D up-sampling convolution layers. In both the encoder and decoder parts, batch normalization and Rectified Liner Unit (ReLU) activation are applied after each convolution layer. Moreover, the kernel size of the 2D convolution layer is $6 \times 6$ while both the stride and padding sizes are 2.

The feature map generation part is constructed by 3D ResNet with seven layers. Each layer consists of two 3D convolution sub-layers. In the first sub-layers, we apply batch normalization and LeakyReLU to the sub-layer while batch normalization is applied to the second sub-layer. In both the sub-layer, the kernel size of the convolution layer is $3 \times 3$. Moreover, to keep the output size of 3D ResNets unchanged, we use padding of $1 \times 1 \times 1$ at each convolution.

The discriminator $D_2$ is a feed-forward network with six 3D down-sampling convolution layers. Batch normalization and ReLU activation are applied after each convolution layer. The kernel size of the convolution layer is $5 \times 5$ while both the stride and padding sizes are 2.

Similar with the training of the base image generator, the multi-focus cell image generator is trained by the 3D version of the WGAN-GP loss function. Practically, the loss function $\mathcal{L}_G^{(2)}$ of the multi-focus cell image generator $G_2$ is described by

$$\mathcal{L}_G^{(2)} = -\mathbb{E}_{V \sim P_g}[D_2(V)]. \tag{4}$$

On the contrary, the loss function $\mathcal{L}_D^{(2)}$ of the discriminator $D_2$ is defined as

$$\mathcal{L}_D^{(2)} = \mathbb{E}_{V \sim P_g}[D_2(V)] - \mathbb{E}_{V^* \sim P_r}[D_2(V^*)] + \lambda_2 \mathbb{E}_{\hat{V} \sim p_{\hat{V}}}[(\|\nabla_{\hat{V}} D_2(\hat{V})\|_2 - 1)^2] \tag{5}$$

where $V^*$ and $V$ are the sequences of real and synthesized multi-focus images of cells. The value of $\lambda_2$ in our method is set to 10. In eq. (5), $\hat{V}$ is calculated by

$$\hat{V} = \epsilon_2 V^* + (1 - \epsilon_2)V \tag{6}$$

where $\epsilon_2$ is a random number follow $U \sim [0, 1]$.

## 3  Experimental results

To verify the applicability of the proposed method, we made experiments of synthesizing multi-focus images of cells. In our experiments, a digital slide scanner (Hamamatsu Photonics: Nanozoomer-XR) is used to acquire WSI of a sample including many cells. WSI consists of 11 multi-focus images of the sample at different focus. Each WSI has $75{,}000 \times 75{,}000$ [pixel] while the spatial resolution of each image in WSI is 0.23 [$\mu$m/pixel]. The multi-focus images of a target cell

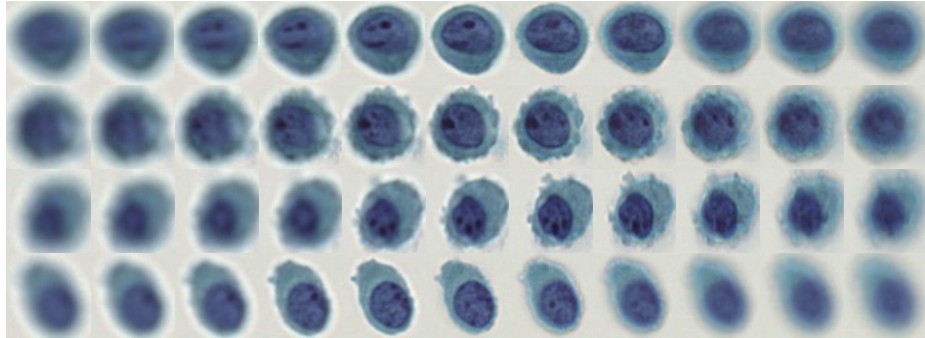

**Fig. 4.** Examples of real multi-focus images of SiHa cells.

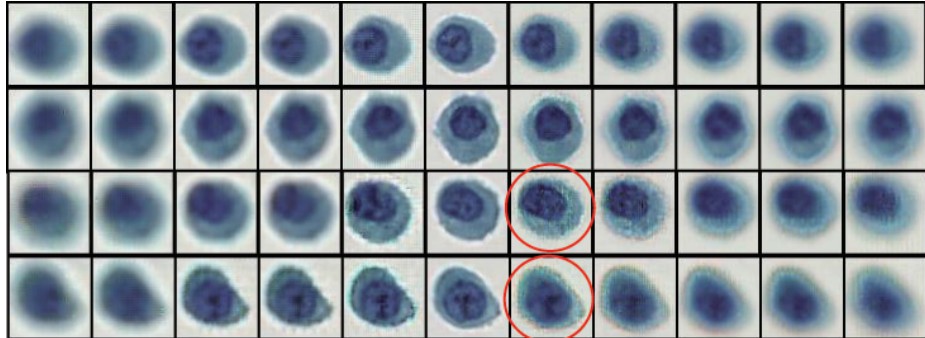

**Fig. 5.** Synthesized multi-focus SiHa images by MI-GAN1.

are extracted automatically from the WSI. The size of each cell image is $64 \times 64$ [pixel]. The proposed method is implemented on a commercial desktop computer (Quadro GP100 16GB and Pytorch framework).

Firstly, we constructed MI-GAN, called MI-GAN1, for generating multi-focus images of SiHa cell which is one of human cervical cancer cell lines. Fig. 4 shows the examples of the real multi-focus images of SiHa cell. the MI-GAN1 construction uses 1,100 images of SiHa cells. In addition, to prevent the proposed system overfitting, we perform data augmentation as follows: 90, 180, and 270 [deg] rotation of the original data, and a mirror flip of the up-down and left-right directions. Finally, MI-GAN1 is constructed by using 6,600 SiHa images.

The proposed method synthesized the sequences of multi-focus SiHa images as shown in Fig. 5. From these figures, the proposed method can generate realistic multi-focus SiHa images compared with real SiHa cell images in Fig. 4. Moreover, the quality of the synthesized cell images is evaluated by some experienced cytotechnologists. We got the comment of the experienced cytotechnologists that the synthesized cell images are very similar with real cell images.

However, MI-GAN1 generates some multi-focus images with low quality. The red circles in Fig. 5 illustrate the example of the low quality cell images obtained

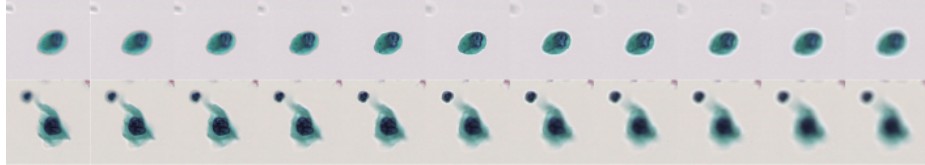

**Fig. 6.** Examples of real multi-focus images of cancer cells.

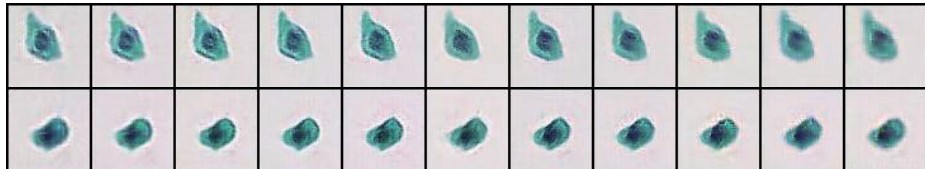

**Fig. 7.** Synthesized multi-focus images of cancer cells by MI-GAN2.

by MI-GAN1. The figures include some noises in the smooth region of cell cytoplasm and unnatural change of the cell nuclear shape between successive images. The solutions for this problem include the improvement of the network architecture and the definition of the loss function.

The second experiment is to construct another MI-GAN, called the MI-GAN2, for generating multi-focus images of real cancer cells (Fig. 6). The number of real cancer cell images is 541 and about half of SiHa cell images used in the MI-GAN1 construction. Therefore, a transfer learning is applied to construct MI-GAN2. Practically, the trained MI-GAN1 using SiHa cell images is used as the initial architecture of MI-GAN2. Moreover, the data augmentation is applied to increase the number of the cancer cell images. Using the cancer cell images, MI-GAN2 is trained to synthesize multi-focus images of cancer cells.

The generated multi-focus images of cancer cells is illustrated in Fig. 7. As with the generation of SiHa cell image, the proposed method reconstructs multi-focus images of cancer cells with acceptable quality compared with real cancer cell images. From the results, the pretrained GAN by using one type of cells is useful to construct GAN for producing another type of cells.

## 4   Conclusion

We propose a GAN-based method, MI-GAN, for synthesizing multi-focus images of cells. The synthesis process using MI-GAN is composed of two phases. In the first phase, from random noise, the proposed method, MI-GAN produces a base cell image which is in focus in the multi-focus images. In the second phase, the MI-GAN produces realistic multi-focus images of the cell considering the relationships between the images acquired at successive focus points. From the experimental results, MI-GAN achieves the good performance to generate realistic multi-focus cell images. One of our future works is to establish metrics

for evaluating the generated multi-focus cell images such as the visual tuning test [14].

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
