# OpenReview forum: "GAN-based Method for Synthesizing Multi-Focus Cell Images"
_MICCAI.org/2019/Workshop/COMPAY — Submitted to COMPAY 2019_

### Official Review · AnonReviewer1 · 2019-07-25
**Interesting concept, but no validation and motivation?**

**Rating:** 4
**Confidence:** 4

**Review:**

Thank you for submitting your manuscript. The paper is well written and understandable.

However, I have one minor and one major concern:

- the authors should elaborate on their motivation and problem statement. Synthesizing multi-focus images of cells is a very specific application in pathology. It would be good to add a few sentences why this is useful, what can be improved with that. The authors talk about an automated cervical cancer screening system, but do not clarify how the proposed generation of multi-focal images plays in with that.

- the major concern is the (basically missing) validation. In the abstract and conclusion, the authors state "good" performance, without any notion of what "good" means. They do not say how many pathologists have looked over how many cells, nor they said anything about training/test splits or whatsoever.
Please think of quantitative results. One idea would be: how big is the error to the real multi-focus images, if the multi-focus cell image generation network starts with the middle image of the real stack?

- Specify how many pathologists looked at how many cells to confirm "good" results. And what does "good" mean? What do pathologists expect from such multi-focal images (e.g. to see nucleoli? shape?) Can they see the same features in the generated ones? What would they do with that downstream?

- why did you use only one WSI, and 1100 nuclei? There should be magnitudes more on a WSI with 75k x 75k px.

- You say the nuclei were detected automatically. How?

- Please explain the figures more in the headlines.

- In Fig. 1, what does MHI stand for?

- Any idea why you have to retrain the model for cancer images? What happens if you just transfer the network w/o finetuning?

---

### Official Review · AnonReviewer2 · 2019-08-02
**Good start that needs more evaluation**

**Rating:** 5
**Confidence:** 4

**Review:**

The paper describes a GAN method for synthesizing multi-focus cell images, with the motivation of subsequently using it to expand training data for cervical cancer screening applications.

The strongest point of the paper is the adaption of GAN methods to the 3D problem of multi-focus images. This is a nice contribution to the computational pathology domain.

One major question mark is, however, not straightened out: Are the synthesized images useful for their purpose? A first sanity check is made in the paper, cytotechnologists express that they are “similar” to real images. But it’s very difficult to know what that statement is worth. One option would have been to let human experts do a formal evaluation, to at least get an idea of how large portion of the synthesized images was clearly off the mark (such as the ones circled in fig 5). But there is no guarantee that such assessments directly translates to the desired enrichment of training data. So in my opinion, the real test is to use the synthesized data in training and see if classification performance benefits from it. In the field, there seems to be mixed experiences of using synthesized data, sometimes it helps training, sometimes it doesn’t.

The paper is mostly very clearly written and easy to follow. I have some trouble grasping the input data sets, though. More clarity is needed about the different data sources, how cell extraction was done etc. This unclarity also makes it difficult to assess experiment 2. Better explanation is needed of what the main challenge of moving from synthesizing SiHa vs real cancer cells is, and why starting with SiHa and then transferring is a good idea. And here the evaluation is completely missing, it is difficult to assess whether the results are successful or not.

All in all, this is a very nice start that surely will lead to good contributions,  but in my view some more work on the evaluation is needed to establish that the method is mature enough.